# Analysis of the Genetic Diversity of Two *Rhopalosiphum* Species from China and Europe Based on Nuclear and Mitochondrial Genes

**DOI:** 10.3390/insects14010057

**Published:** 2023-01-06

**Authors:** Jianqing Guo, Jing Li, Sebastien Massart, Kanglai He, Frédéric Francis, Zhenying Wang

**Affiliations:** 1College of Agriculture and Forestry, Hebei North University, Zhangjiakou 075000, China; 2State Key Laboratory for Biology of Plant Diseases and Insect Pests, Institute of Plant Protection, Chinese Academy of Agricultural Sciences, Beijing 100193, China; 3Department of Functional and Evolutionary Entomology, Gembloux Agro-Bio Tech, University of Liège, Passage des Déportés 2, B-5030 Gembloux, Belgium; 4School of Biological and Environmental Engineering, Xi’an University, No. 1 Keji Six Road, Xi’an 710065, China

**Keywords:** EF-1α, genetic differentiation, geographical region, mtDNA, *Rhopalosiphum maidis*, *Rhopalosiphum padi*

## Abstract

**Simple Summary:**

*Rhopalosiphum padi* and *Rhopalosiphum maidis* are two common phloem-sucking pests on maize which can even transmit viruses (i.e., maize dwarf mosaic virus, MDMV) leading to severe yield losses of maize. Population genetic studies provide information about how much ecological and genetic divergence has occurred among many near and distant populations. We investigated the genetic diversity of both *Rhopalosiphum* aphids collected from maize in China and Europe, and we found that different populations of *R. maidis* showed low genetic variation, indicating a high level of gene flow of both nuclear and mitochondrial genes in this aphid. However, the mitochondrial gene of *R. padi* exhibited obvious genetic differentiation between Chinese samples and European samples. In conclusion, the domestic populations of both *R. padi* and *R. maidis* showed low genetic diversity, and the long distance between China and Europe may interrupt the gene exchange of aphids.

**Abstract:**

Population genetic studies can reveal clues about the evolution of adaptive strategies of aphid species in agroecosystems and demonstrate the influence of environmental factors on the genetic diversity and gene flow among aphid populations. To investigate the genetic diversity of two *Rhopalosiphum* aphid species from different geographical regions, 32 populations (n = 535) of the bird cherry-oat aphid (*Rhopalosiphum padi* Linnaeus) and 38 populations (n = 808) of the corn leaf aphid (*Rhopalosiphum maidis* Fitch) from China and Europe were analyzed using one nuclear (elongation factor-1 alpha) and two mitochondrial (cytochrome oxidase I and II) genes. Based on the COI-COII sequencing, two obvious clades between Chinese and European populations and a low level of gene flow (Nm = 0.15) were detected in *R. padi*, while no geographical-associated genetic variation was found for EF-1α in this species. All genes in *R. maidis* had low genetic variation, indicating a high level of gene flow (Nm = 5.31 of COI-COII and Nm = 2.89 of EF-1α). Based on the mitochondrial result of *R. padi*, we concluded that the long distance between China and Europe may be interrupting the gene flow. The discordant results of nuclear gene analyses in *R. padi* may be due to the slower evolution of nuclear genes compared to mitochondrial genes. The gene exchange may occur gradually with the potential for continuous migration of the aphid. This study facilitates the design of control strategies for these pests.

## 1. Introduction

Intraspecific genetic diversity provides the basis for investigating the evolutionary history of species and offers an opportunity to document the basic level of biodiversity within and among populations [1]. Hence, population genetics have highlighted the importance of studying the molecular variability within species for population genetics. In addition to the life cycle of aphids [2] and host plant species [3,4], the geographical isolation of aphid populations may also generate a genetic structure resulting from both drift and selection under different environmental conditions [5]. For instance, a significant correlation between aphid populations and the geographical regions of the bird cherry-oat aphid (*Rhopalosiphum padi* Linnaeus; Hemiptera: Aphididae) was detected in Iran [6]. Seven populations of the grain aphid (*Sitobion avenae* Fabricius) collected from different geographical regions in China showed high genetic diversity [7]. Furthermore, the genetic variation of the sorghum aphid (*Melanaphis sacchari* Zehntner) appeared to be more strongly influenced by geography than by host plants [8]. In addition, the *S. avenae* populations exhibited life-history divergence and local adaptation in northern and southern areas separated by the Qinling Mountains of China [9]. Geographical barriers may lead to genetic diversity and interrupt the gene flow among populations within species [6]. By contrast, genetic differentiation among populations can be homogenized by migration, resulting in an increase in the gene flow [10].

The corn leaf aphid (*Rhopalosiphum maidis* Fitch; Hemiptera: Aphididae) and *R. padi* are the two common phloem-sucking pests on maize (*Zea mays* Linnaeus), one of the major crops in China. In addition, *R. maidis* can also feed on broomcorn (*Sorghum bicolor* L.), wheat (*Triticum aestivum* L.), barley (*Hordeum vulgare* L.), and goose grass (*Eleusine indica* L.) [11], while *R. padi* often attacks peach (*Prunus persica* L.), plum (*Prunus salicina* Lindl.), and bird cherry (*Prunus padus* L.) cyclically besides Gramineae [12]. These pests are widely distributed around the world, including in Europe [13,14,15,16], America [17], Canada [18,19], Australia [20], North Africa [21], New Zealand [22,23], India [24], Egypt [25], and China [11,26,27]. Both *Rhopalosiphum* species can transmit viruses including the maize dwarf mosaic virus (MDMV) and the barley yellow dwarf virus (BYDV) [28,29,30], leading to a decreased quality and yield of maize, which in turn may cause serious economic damage. Hence, it is necessary to understand the genetic diversity and distribution of these two aphids as they may provide clues for understanding their migration and expansion, which in turn may facilitate their management. 

Genetic diversity is affected by factors including the life cycle of aphids [26], the host plant species [31,32], and geographical conditions [33,34]. Evidence showed that *R. maidis* generally lacks sexual reproduction [35], and it often overwinters with parthenogenetic adults or nymphs in China [36]. Furthermore, little is known about the genetic divergence of this aphid. On the other hand, most genetic studies of *R. padi* have focused on its life cycles or combination of life cycles and geographical locations [2,13,26,37,38,39,40], rather than the geographical locations alone [15,20]. There are four types of reproductive modes of *R. padi* in China [41]: (1) cyclical parthenogenesis; (2) obligate parthenogenesis; (3) mixture of cyclical parthenogenesis and obligate parthenogenesis; (4) obligate parthenogenesis with male production. In addition, markers such as RAPD [23], SSR [6,26,42], ISSRs [25], and mtDNA (mitochondrial DNA) [20,38], as well as the genotyping-by-sequencing approach [39], have been applied to reveal genetic variations of *R. padi*. So far, only two studies on the genetic differentiation of *R. maidis* based on allozymes [43], mtDNA, and chromosome number [21] have been published. 

According to previous research on *R. padi*, it is known that: (1) There is a significant genetic difference between reproductive modes [2,37,38,40]. Cyclically parthenogenetic populations showed little regional genetic diversity, while considerable geographical differentiation existed among obligate parthenogenesis populations [37,38]; however, two geographical genetic clusters were identified from the samples of England which were mostly cyclical parthenogenetic [39]. (2) The genetic differentiation among various geographical localities is generally low in New Zealand [22,23], some European countries [16,44], Canada [19], and Australia [20]. (3) The distribution of some *R. padi* clones can be driven as well by longitudinal clines [42]. As for *R. maidis*, based on the scarce reports available, Steiner et al. found that the northern populations of *R. maidis* in North America lacked heterozygosity and were genetically diverse compared with the populations from southern regions [43]. However, Simon et al. reported that genetic differentiation among various populations of *R. maidis* was related to the host plants rather than geographical localities [21]. 

Among all the molecular markers for genetic variation research on insects, mtDNA is considered a useful and efficient tool because of the maternal inheritance, lack of recombination, and high mutation rate [45]. Mitochondrial cytochrome oxidase I (COI) and II (COII) have been repeatedly employed to assess host- or geographical location-associated genetic variation in aphids [46,47,48,49,50]. Nuclear regions such as the elongation factor-1 alpha (EF-1α) gene have also been used in many genetic studies on aphids [51,52,53,54].

So far, there is only one report on the genetic divergence of *R. padi* collected from maize [42]. Moreover, no studies on the genetic diversity of *R. maidis* from maize are available. Therefore, this study was conducted to describe for the first time the geographical genetic differentiation among populations of these two *Rhopalosiphum* species on maize from most maize-sowing areas in China as well as several European countries, based on the COI, COII, and EF-1α genes. We also investigated whether the genetic diversity of *R. padi* and *R. maidis* was influenced by geographical variables of the maize-producing regions, which may provide the basis for the management of these aphids.

## 2. Materials and Methods

### 2.1. Aphid Collection and DNA Preparation

A total of 1343 aphid samples (535 *R. padi* of 32 sites and 808 *R. maidis* of 38 sites) were collected from maize in China and Europe from 2014 to 2016. Based on the climate and maize cultivation practices, the maize cultivation areas in China are mainly divided into five regions defined as the north spring maize region (NS), Huanghuaihai summer maize region (HS), southeast hilly maize region (SEH), northwest inland maize region (NWI), and southwest hilly maize region (SWH), respectively (Figure 1 and Figure 2). Furthermore, the specimens collected from Europe in our study were named as European countries maize region (EUR) for *R. padi* and maize region in France (FRA) for *R. maidis*. Collection information and localities are listed in Appendix A and Figure 1 and Figure 2. The distance between two aphid sampling plants within one sampling locality was at least 10 m to reduce the probability of sampling the same aphid genotype repeatedly. Two or more aphids were collected for each sample in case the DNA extraction from one individual failed. All specimens were preserved in absolute ethanol and stored at −20 °C before the molecular analysis. Genomic DNA was extracted from a single aphid using TEN (10 mM Tris-HCl pH = 8, 2 mM EDTA pH = 8, 0.4 M NaCl), 20% SDS, and 5 M NaCl solution according to the salting-out method [55]. Then, 20–30 µL TE buffer was used to dissolve the precipitated DNA, and then the DNA samples were kept at −20 °C for further use.

### 2.2. Gene-Specific PCR and DNA Sequencing

Partial sequences of COI, COII, and EF-1α genes were used in this study for phylogenies analyses. Primers were designed utilizing Primer Premier 5.0 (http://www.premierbiosoft.com/ (accessed on 1 January 2020)) with the sequences downloaded from Genbank (Appendix A). A PCR (polymerase chain reaction) was performed in a 25 μL volume containing 1.5 μL (around 40 ng) of template DNA, 400 nM of each primer, and 12.5 μL of 2 × GoTaq ^®^ Colorless PCR MasterMix (Promega, Madison, WI, USA). In addition, BSA (bovine serum albumin) (10 mg/mL) with 1% of the total volume was added to the reaction system for all the genes except for the EF-1α of *R. maidis* in order to improve amplification results. The PCR was implemented under the following conditions: initial denaturation at 94 °C for 3 min; 35 amplification cycles of 94 °C for 30 s, annealing temperature (Appendix A) for 35 s, and 72 °C for 35 s; and a final extension at 72 °C for 10 min. PCR products were sent for sequencing (Sangon, Beijing, China) with the sense primers. 

### 2.3. Data Analysis

The obtained sequences were verified via BLAST (http://blast.ncbi.nlm.nih.gov/Blast.cgi (accessed on 1 January 2020)) and the chromatograms were checked for the nucleotide variations. The unreliable terminal sequences were cut by MEGA v6 software after multiple alignment. The haplotype numbers (Ha), haplotype diversity (Hd), number of polymorphic sites (S), nucleotide diversity (Pi), Tajima’s D neutrality test [56], Fu’s Fs [57], and overall gene flow values (Nm: N, the effective population size; m, migration rate per generation) were calculated using the DnaSP v5.10 software [58]. Haplotype networks were constructed using Network v4.6 [59]. The phylogenetic analyses were conducted using the maximum likelihood (ML) methods with bootstrap support (1000 replicates) [60] by MEGA v6 software, and a Tamura-Nei model was used to conduct the phylogenetic analyses. COI and COII were concatenated for both *R. padi* and *R. maidis*. COI, COII, and EF-1α sequences of the greenbug (*Schizaphis graminum* Rondani) downloaded from NCBI (http://www.ncbi.nlm.nih.gov/ (accessed on 1 January 2020)) were used as the outgroup for *R. padi* and *R. maidis*. Molecular variance (AMOVA) analyses of pairwise F_ST,_ Φ_ST_ (genetic differentiation), Nm values between geographic populations, as well as five geographical groups (i.e., five maize-sowing regions) were performed using ARLEQUIN v3.5 [61] with 1000 permutations. The analyses of Nm values were based on assumptions of the population size and structure (i.e., the island model): the equilibrium between migration and drift, no selection, no mutation, stable population size, and the same size of subpopulations [62]. The pairwise Nm was calculated based on the pairwise F_ST_ and Φ_ST_ values according to the equilibrium between migration and drift: Nm = (1 − Φ_ST_)/2Φ_ST_ for mitochondrial genes and Nm = (1 − F_ST_)/4F_ST_ for EF-1α. In addition, the pairwise Φ_ST_ of mtDNA was estimated as the ratio of genetic differentiation between populations weighted by the degree of within-population differentiation [63]. The observed and expected distributions of the number of pairwise genetic differences (mismatch distributions) were performed using DnaSP v5.10 software, and the sum of squared deviation (SSD) as well as Harpending’s Raggedness index were calculated under the sudden expansion model by ARLEQUIN v3.5. The influence of geographic distance on population genetic diversity was performed by assuming a simple island model in an isolation by distance (IBD) analysis [64]. Mantel tests were carried out using GenAlEx 6.502 [65], between the pairwise genetic distances of F_ST_/(1 − F_ST_) for EF-1α and Φ_ST_/(1 − Φ_ST_) for mtDNA and the logarithms of geographical distances (km) between all collecting locations.

## 3. Results

### 3.1. Genetic Diversity and Sequence Variation

We obtained high quality sequences of 496 nt, 499 nt, and 672 nt of COI, COII, and EF-1α, respectively, for *R. padi*, as well as 568 nt, 497 nt, and 668 nt of COI, COII, and EF-1α, respectively, for *R. maidis*. An analysis of haplotype diversity (Table 1) showed that only 12 and 7 haplotypes were detected from the combined COI-COII (1065 nt) and EF-1α (668 nt) sequences, respectively, among 808 *R. maidis* individuals. Furthermore, 29 and 32 haplotypes were detected from the combined COI-COII (995 nt) and EF-1α (672 nt) genes, respectively, among 535 *R. padi* individuals. All haplotype sequences were submitted to GenBank with the accession numbers of KY612517 to KY612597. 

The haplotype diversity (Hd) of *R. padi* was 0.663 for the combined COI-COII, being higher than that of *R. maidis* (0.227). In addition, the Hd values for the EF-1α gene of *R. padi* and *R. maidis* were 0.507 and 0.548, respectively. The nucleotide diversity (Pi) of *R. padi* was 3.52 × 10^−3^ for the combined COI-COII and 1.04 × 10^−3^ for the EF-1α, both of which were higher than the Pi of *R. maidis* (0.22 × 10^−3^) for the combined COI-COII and EF-1α (0.89 × 10^−3^). Neutrality tests were conducted using Tajima’s D and Fu’s Fs statistics, and all sequences were included as one group for each species (Table 1). Tajima’s D was significantly negative (*p* < 0.05) for the EF-1α of *R. padi* and the combined COI-COII of *R. maidis*. Nevertheless, neutrality tests using Tajima’s D values for each population showed that only four populations (TH, TMT, YuL, and ZJK) of EF-1α in *R. padi* and six populations (CQ, GZ, NC, SJZ, XX, and YuL) of the combined COI-COII in *R. maidis* were significantly negative (*p* < 0.05) (Appendix A). Additionally, the results of Fu’s Fs test exhibited that 2 populations (DN, TMT) of the combined COI-COII in *R. padi*, 11 populations (GM, DN, FR, HBD, LF, QT, TH, TL, TMT, WF, and ZJK) of EF-1α in *R. padi*, and 11 populations (GZ, HS, LF, MZ, NC, SJZ, SZ, YaL, XX, YuL, and ZY) of the combined COI-COII in *R. maidis* were significantly negative (*p* < 0.05) (Appendix A). A low level of overall gene flow (Nm = 0.15) was detected for the combined COI-COII in *R. padi*, whereas a higher gene flow (Nm = 1.86) was found for EF-1α in *R. padi*, as well as both sequences in *R. maidis* (Nm = 5.31 of the combined COI-COII and Nm = 2.89 of EF-1α).

The polymorphic nucleotide sites of the combined COI-COII and EF-1α genes in the two *Rhopalosiphum* aphids are illustrated in Appendix A. These haplotypes revealed 32 and 22 polymorphic sites for the combined COI-COII and EF-1α genes of *R. padi*, and 11 and 4 polymorphic sites for the combined COI-COII and EF-1α sequences of *R. maidis*, respectively. In total, 84.1% of the polymorphic sites (58/69) were transitions, of which 40 sites were T/C transitions and 18 were G/A transitions. In addition, approximately 14.5% (10/69) transversions were found, six of which were T/A transversions and the other four were G/T transversions. One polymorphic site contained both a transition and a transversion at 634 nt for the combined COI-COII sequence of *R. padi*. No insertions or deletions were detected for all sequences.

### 3.2. Genetic Structure of the Populations

An AMOVA analysis was conducted based on both one group (considering all populations as one group) and five groups (the populations were divided into five groups based on the maize-sowing regions) (Appendix A, Figure 1 and Figure 2). The results of the one-group analysis (Appendix A) revealed that the main genetic variation was found within populations. In particular, the percentage of variation reached 93.48% for the combined COI-COII and 95.44% for the EF-1α of *R. maidis*. Similarly, the percentages of variation within populations were also high for *R. padi* (61.91% of the combined COI-COII and 83.33% of EF-1α). Moreover, the genetic distance (F_ST_) of *R. padi* (0.167 for EF-1α and 0.381 for the combined COI-COII) had no significant difference with *R. maidis* (0.046 for EF-1α and 0.065 for the combined COI-COII). When the populations were geographically divided into five groups based on the maize-sowing regions, the genetic variation within populations was predominant (Appendix A). Furthermore, a higher percentage of variation within groups than among groups was observed for all sequences, except for the combined COI-COII of *R. padi*, where 31.00% of the variation was found among groups, while the percentage of variation within groups was 12.45%.

The pairwise F_ST_ values of EF-1α and Φ_ST_ values of mtDNA between geographical populations are displayed in Appendix A. As for *R. padi*, the 496 pairwise values among the 32 populations ranged from −0.266 (SYS and HEB) to 1 (MS and JNi; MS and SZ) of Φ_ST_ values for the combined COI-COII (Appendix A), and −0.248 (SYS and HBD) to 0.730 (SYS and JNi) of F_ST_ values for EF-1α (Appendix A). We found that 64.5% (n = 320) of the Φ_ST_ value for the combined COI-COII and 37.1% (n = 184) of F_ST_ values for EF-1α showed statistically significant genetic differentiation (*p* < 0.05). In general, most pairwise Φ_ST_ values of *R. padi* for populations from different regions were higher than those for populations from the same region. For instance, higher pairwise Φ_ST_ values for the combined COI-COII were found between the EUR population and populations from other regions compared to those between EUR populations. However, most F_ST_ values of the EF-1α gene were low even between populations from different regions. The Nm values of *R. padi* ranged from 0 (MS and JNi, MS, and SZ) to 1785.214 (LF and BJ) for the combined COI-COII (Appendix A), and 0.092 (SYS and JNi) to 151.469 (LF and DN) for EF-1α (Appendix A). In addition, we detected 30.2% (n = 150) for the combined COI-COII and 23.4% (n = 116) for EF-1α of the Nm values were smaller than 1.

Regarding the 38 populations of *R. maidis*, the 703 pairs of Φ_ST_ values varied from −0.018 (BJ and TL) to 0.318 (ZJK and DZ) for the combined COI-COII (Appendix A), and the F_ST_ values for EF-1α ranged from −0.107 (JNa and TMT) to 0.625 (JNa and YC) (Appendix A). The proportion of significant values (*p* < 0.05) was 16.2% (n = 114) of the Φ_ST_ for the combined COI-COII, and 16.1% (n = 113) of the F_ST_ for the EF-1α gene. Moreover, some populations, such as YC (SWH region, EF-1α, *R. maidis*), DZ (HS region, combined COI-COII, *R. maidis*), and MS (SWH region, both the combined COI-COII and EF-1α, *R.padi*), exhibited high pairwise values for populations both within and outside of the same region. The 703 Nm values of *R. maidis* varied from 1.071 (ZJK and DZ) to 6249.500 (ZY and NC) for the combined COI-COII (Appendix A) and 0.150 (YC and JNa) to 942 (MZ and HS) for EF-1α (Appendix A). In addition, none of the Nm values for the combined COI-COII and 10.1% (n = 71) of EF-1α of the total 703 Nm values were under 1, implying a high level of gene flow among *R. maidis* populations.

Genetic isolation by geographic distance was not significant for both mtDNA and EF-1α of *R. padi* (Mantel test, *p* = 0.1 for mtDNA and *p* = 0.21 for EF-1α, Figure 3a,b). As for *R. maidis*, the mitochondrial haplotype data did not reveal any significant IBD for Φ_ST_ values (Mantel test, *p* = 0.24, Figure 3c), whereas a modest IBD was discovered in EF-1α for the F_ST_ values (Mantel test, *p* = 0.02, *R*^2^ = 0.10, Figure 3d).

### 3.3. Haplotype Phylogeny and Network

The phylogenetic analysis of haplotypes is shown in Figure 4 and Figure 5. The 29 haplotypes of the combined COI-COII genes in *R. padi* were divided into two clusters (Figure 4a), of which all haplotypes belonging to one cluster were observed in aphids collected from China. By contrast, the haplotypes belonging to the other cluster were European samples, except for three individuals from the NWI region of China. A similar result was obtained by a network construction, which exhibited two distinct haplotype clusters from Chinese and European samples (Figure 6a). In addition, most of the individuals (109/121 specimens) from the HS region were distributed in HRp1 (haplotype 1 of *R. padi*), 30/53 specimens collected from the SWH region were distributed in HRp2, and 20/41 samples from the NWI region were distributed in HRp3. However, no significant regional distribution was observed for the 32 haplotypes of the EF-1α gene in *R. padi*, using both the phylogenetic tree (Figure 4b) and the network results (Figure 6b). This may be explained by the lack of genetic diversity in the EF-1α gene of *R. padi* because most haplotypes were separated by only one nucleotide substitution (Appendix A). 

Regarding *R. maidis*, both the phylogenetic (Figure 5a) and network (Figure 7a) results of the 12 haplotypes in the combined COI-COII genes showed low genetic diversity. Furthermore, among the seven haplotypes detected in the EF-1α gene of *R. maidis* (Figure 5b and Figure 7b), HRm1 (haplotype 1 of *R. maidis*; 384/808 specimens) and HRm4 (384/808 specimens) were the dominant haplotypes based on the network results (Figure 7b). In addition, all samples assigned to HRm6 and HRm7 were collected from France, while three specimens from this region were distributed in HRm1, and four specimens, in HRm4.

### 3.4. Demographic History of R. padi and R. maidis

Mismatch distribution analyses of mitochondrial and nuclear genes were performed to examine the historical demographic expansion of *R. maidis* and *R. padi* (Figure 8). The values of Tajima’s D were not significant for the EF-1α of *R. maidis* (Figure 8b) and mtDNA from *R. padi* (Figure 8c) (*p* > 0.10) (Table 1). However, no multiple peaks could be observed in Figure 8a (mtDNA of *R. maidis*) and 8d (EF-1α of *R. padi*), and the values of Tajima’s D were significant for the mtDNA of *R. maidis* and EF-1α from *R. padi* (*p* < 0.05) (Table 1). In addition, the network of mtDNA from *R. maidis* (Figure 7a) showed a stellate radiation, which suggests that these aphids have experienced population fluctuation over time.

## 4. Discussion

### 4.1. Genetic Diversity and Sequence Variation

This is the first study of the genetic diversity of *R. padi* collected from maize in most of the maize-growing regions in China as well as Europe. Knowledge about the genetic diversity of *R. maidis* has been so far limited to two studies based on an allozyme analysis of North American populations, which showed regional differentiation [43], and Canadian populations, which demonstrated low genetic diversity [21]. Our results revealed that *R. maidis* exhibited low genetic differentiation among populations. The genetic diversity of *R. padi* was overall higher than that of *R. maidis* particularly for the combined COI-COII, which showed the highest Hd values (0.663) and lowest overall Nm values (0.15). However, low geographic genetic diversity was reported for *R. padi* in Australia, based on the analysis of COI and microsatellites [20]. 

### 4.2. Genetic Structure of the Population

According to the AMOVA results, the highest proportion of the genetic variability observed in the mitochondrial and nuclear gene sequences was explained by within-population variation in both aphid species. The genetic variance within populations seems to contribute much more than that among populations for various aphid species including *S. avenae* [7,66], the woolly apple aphid (*Eriosoma lanigerum* Hausmann) [67], and *R. padi* [37]. For *R. padi,* the variations of the mitochondrial gene among the groups in different maize-sowing regions were significantly larger than those within groups, which is likely explained by the geographical division between Chinese and European specimens. Furthermore, different maize varieties between China and Europe may also contribute to the genetic differentiation of *R. padi* since the races of host plants act as a trigger for speciation of phytophagous insects [68]. Nevertheless, for *R. maidis,* both genes showed greater variation within groups than among groups because of the low genetic diversity in China and the use of only one collection site in France, which may not be representative enough for a region outside of China.

The pairwise F_ST,_ Φ_ST_, and Nm values exhibited a low level of significant genetic differentiation but a high frequency of gene flow for *R. maidis*. Less than 20% of the significant genetic differentiation (*p* < 0.05) for both the combined COI-COII and EF-1α, and none of the Nm values for the combined COI-COII were less than 1, revealing that there was very little genetic differentiation among *R. maidis* populations [69]. Hence, no obvious genetic structure could be formed as illustrated in the network results. Obligate parthenogenesis of *R. maidis* reduces the chance of gene recombination, and the parthenogenetic populations show less allelic polymorphism compared with sexual reproduction [37], which may contribute to the low genetic diversity. Moreover, for *R. padi*, the results of mitochondrial and nuclear gene analyses were to some extent discordant. The proportion of significant genetic differentiation for pairwise Φ_ST_ values for the combined COI-COII (64.5%) was twice as high as that of the F_ST_ values for EF-1α (37.1%) (*p* < 0.05). The discordance between nuclear and mtDNA phylogeny could be explained by the higher rate of evolution of the mitochondrial genome compared to the nuclear genome due to its haploid nature and lack of recombination [70]. In addition, gene flow resulting from the continuous migrations of aphids could interrupt the formation of genetic differentiation. Evidence has proven that *R. padi* [71,72,73] and *R. maidis* [17,18,43] are migratory pests, and insects with small body sizes, such as aphids, often migrate for certain distances carried by the wind [74], which consequently homogenizes the genetic composition.

### 4.3. Haplotype Phylogeny and Network

The phylogeny and network analyses of haplotypes produced consistent results. The two distinct clades of the combined COI-COII (Figure 6a) in *R. padi* revealed that the substantial geographical distance between China and European countries may result in distinct aphid lineages (i.e., subspecies) based on geographical differentiation. By contrast, low genetic diversity has often been detected in insects collected from a small geographical range (e.g., within a country) [49,75,76,77]. However, most specimens from the HS region were distributed in HRp1, and more than half of the specimens from the SWH region and half from the NWI region were distributed in HRp2 and HRp3, respectively, of the mtDNA in *R. padi*. This result indicates that there exists some genetic differentiation within China despite the frequent gene exchange. Furthermore, the sampling sites in Europe (temperate maritime climate) have mild winters, whereas the NWI region (temperate continental area with arid climate) and HS region (temperate monsoon climate with high temperature and rainy summer) have cold winters, which may affect the population genetic diversity of *R. padi* [40]. Regarding the European clade, we detected eight haplotypes not present in Chinese populations. Nevertheless, no obvious genetic diversity was found among the eight haplotypes, which may be the result of the few collection sites (four sites) from the European region because the longitudinal cline-related distribution of *R. padi* in France was described in a previous study [42]. In addition, 3 out of 29 *R. padi* from the HRp6 (in the European clade) of mtDNA were collected from the NWI region of China. We speculate that three possibilities may explain this phenomenon: (1) European *R. padi* might have been passively carried to the NWI region through agricultural products and then colonized the region; (2) a spontaneous mutation occurred in Chinese *R. padi* which resulted in HRp6; (3) a gene exchange might have occurred indirectly through continuous migration across countries between China and Europe, which needs further research. By contrast, the genetic diversity of EF-1α in *R. padi* was low (Figure 6b). The mutation rate of mtDNA is significantly higher than that of nuclear DNA [70], which may explain the discordant results. Furthermore, the genetic variation of EF-1α in *R. padi* may not be related to geographical factors in our case. The HRp1 of EF-1α in *R. padi* can be considered a relatively old haplotype because it has diverged to many other haplotypes. The life cycle can also affect the gene mutations in aphids. A study using mtDNA as a marker demonstrated that *R. padi* has an incomplete life cycle (i.e., exists as an obligate parthenogenetic population) and shows the haplotype I, whereas cyclical parthenogenesis populations exhibit distinct haplotypes [16]. Likewise, a microsatellite marker revealed substantial genetic differences between obligate parthenogenetic and cyclically parthenogenetic populations of *R. padi* collected from wheat [26]. Furthermore, longitudinal clines were reported to affect the distribution of *R. padi* clones in France [42]. The geographical localities combined with the life cycle and the host plants of different *R. padi* populations are objects for further study.

Despite the HRm6 and HRm7 of EF-1α being independent haplotypes from France, some French *R. maidis* specimens were classified in HRm1 and HRm4. In summary, low genetic diversity was detected for both genes in *R. maidis*. However, three regional clusters of *R. maidis* were found in North America through an allozyme analysis [43]. Simon et al. reported the low genetic diversity of *R. maidis* from North America, Europe, and North Africa determined using several methods, except for the differentiation of the chromosome number in relation to the host plant species [21]. The genetic variation can be determined by several factors including gene flow, natural selection, host range, time since separation, and migration [77]. In our study, the low genetic variation of *R. maidis* could be caused by the limited host plant (only maize) and migration of this aphid (both the combined COI-COII and EF-1α showed a high level of gene flow). The formation of geographically isolated clones can be inhibited by frequent gene exchange [78]. Moreover, the limited number (n = 16) of *R. maidis* specimens from France may be insufficient to represent the real situation in Europe. Thus, more samples from a larger area in Europe are needed for obtaining more accurate results in subsequent studies.

In conclusion, geographical distance may play a crucial role in genetic diversity based on the distinct Chinese and European clades revealed by the analysis of the mtDNA of *R. padi*. However, the geographical genetic variation may be counteracted gradually by active or passive migration of the aphids. No correlation was found between the genetic diversity and geographical distance for EF-1α in *R. padi*. On the other hand, both mitochondrial and nuclear genes in *R. maidis* showed low genetic variation, which could be the reason why *R. maidis* exhibited low genetic differentiation. Moreover, the low genetic variation of both genes with a high level of gene flow suggests that *R. maidis* may migrate frequently in terms of its geographical range in China. Our study demonstrated the genetic variation of two *Rhopalosiphum* aphids based on both mitochondrial and nuclear gene analyses, focused on the geographical factor, and found a high level of gene flow, particularly among the collection sites in China. Nevertheless, other molecular markers combined with mitochondrial and nuclear genes, as well as the combination of different factors such as geographical distance, host plants, and the life cycle of these aphids may reveal more accurate results, which need further verification. The results reported here are beneficial for understanding the dispersal process of these pests and developing control mechanisms against them.

## Figures and Tables

**Figure 1 insects-14-00057-f001:**
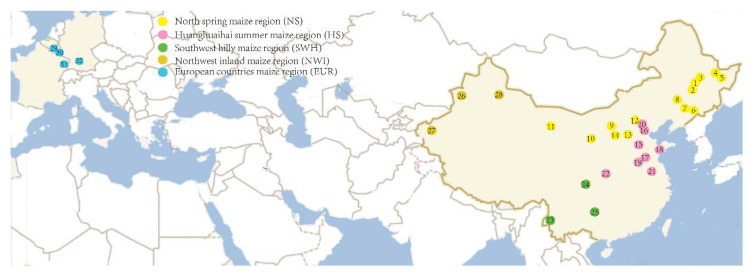
Sampling locations of *R. padi* in China and Europe. Numbers on the map correspond to locality numbers in Appendix A: 1, HEB; 2, HBP; 3, HBD; 4, HG; 5, SYS; 6, TH; 7, SY; 8, TL; 9, TMT; 10, NX; 11, ZY; 12, ZJK; 13, XZ; 14, YuL; 15, HD; 16, LF; 17, JNi; 18, WF; 19, XX; 20, BJ; 21, SZ; 22, YaL; 23, MS; 24, MZ; 25, GY; 26, YN; 27, KS; 28, QT; 29, DN; 30, LSB; 31, FR; 32, GM. Various colors represent different maize-sowing regions.

**Figure 2 insects-14-00057-f002:**
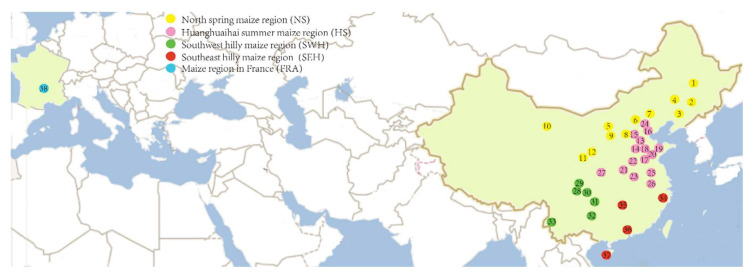
Sampling locations of *R. maidis* in China and France. Numbers on the map correspond to locality numbers in Appendix A: 1, HEB; 2, GZL; 3, SY; 4, TL; 5, TMT; 6, ZJK; 7, LP; 8, XZ; 9, YuL; 10, ZY; 11, TS; 12, PL; 13, HS; 14, HD; 15, SJZ; 16, LF; 17, JNi; 18, DZ; 19, WF; 20, JNa; 21, LY; 22, XX; 23, LH; 24, BJ; 25, SZ; 26, HF; 27, YaL; 28, XD; 29, MZ; 30, NC; 31, CQ; 32, GY; 33, MS; 34, DY; 35, CS; 36, GZ; 37, YC; 38, FR. Various colors represent different maize-sowing regions.

**Figure 3 insects-14-00057-f003:**
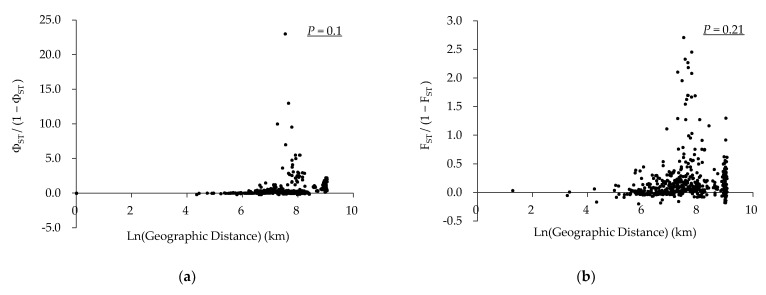
Genetic isolation by geographic distance (km) among different populations of two *Rhopalosiphum* aphids. (**a**): Combined COI and COII genes of *R. padi*; (**b**): EF-1α gene of *R. padi*; (**c**): combined COI and COII genes of *R. maidis*; (**d**): EF-1α gene of *R. maidis*. Significance of the correlation between regression data was estimated from a Mantel test.

**Figure 4 insects-14-00057-f004:**
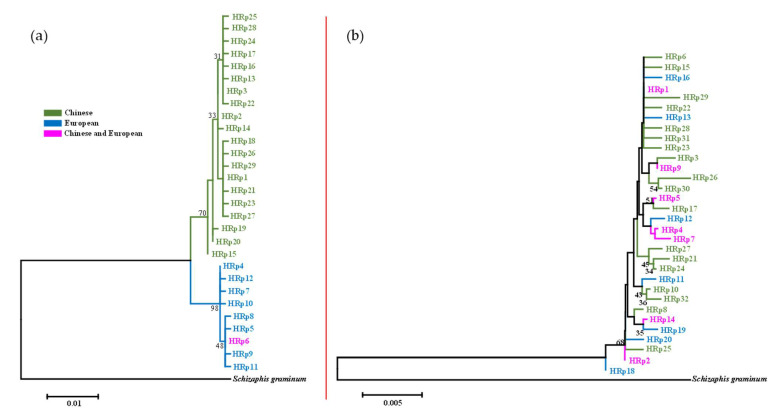
Maximum likelihood (ML) tree showing phylogenetic relationships of *R. padi* with bootstrap support (1000 replicates). (**a**): The 29 haplotypes of combined COI and COII genes (995 nt); (**b**): 32 haplotypes of EF-1α gene (672 nt). HRp: haplotypes of *R. padi*. *Schizaphis graminum* (accession number: HQ392586 for COI, U36751 for COII, and AY219720 for EF-1α) was used as the outgroup insect.

**Figure 5 insects-14-00057-f005:**
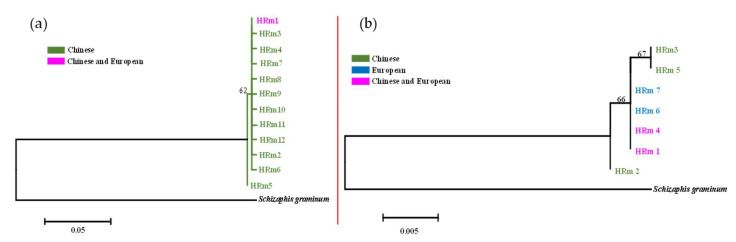
Maximum likelihood (ML) tree showing phylogenetic relationships of *R. maidis* with bootstrap support (1000 replicates). (**a**): The 12 haplotypes of combined COI and COII genes (1065 nt); (**b**): 7 haplotypes of EF-1α gene (668 nt). HRm: haplotypes of *R. maidis*. *Schizaphis graminum* (accession number: HQ392586 for COI, U36751 for COII, and AY219720 for EF-1α) was used as outgroup insect.

**Figure 6 insects-14-00057-f006:**
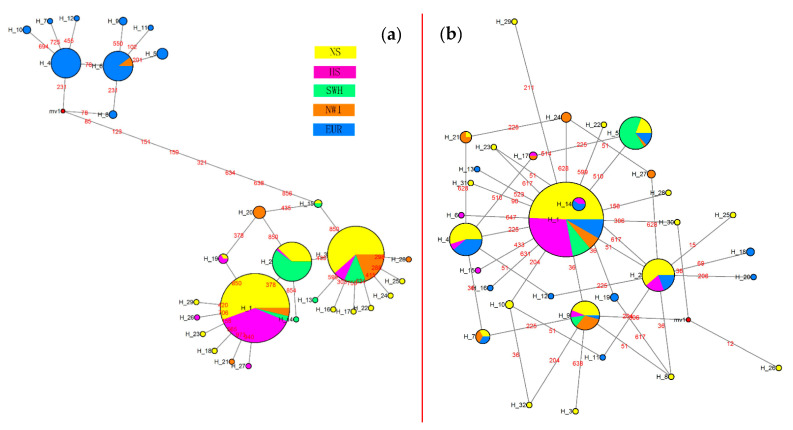
Haplotype network of *R. padi* in China and Europe countries. (**a**): The combined COI and COII genes, H_1 to H_29 represent HRp1 to HRp29; (**b**): the EF-1α gene, H_1 to H_32 represent HRp1 to HRp32. The area of the circles represents the number of individuals sharing the haplotypes. The red numbers indicate the nucleotide positions of polymorphic sites between two haplotypes. NS, north spring maize region in China; HS, Huanghuaihai summer maize region in China; SWH, southwest hilly maize region in China; NWI, northwest inland maize region in China; EUR, European countries maize region.

**Figure 7 insects-14-00057-f007:**
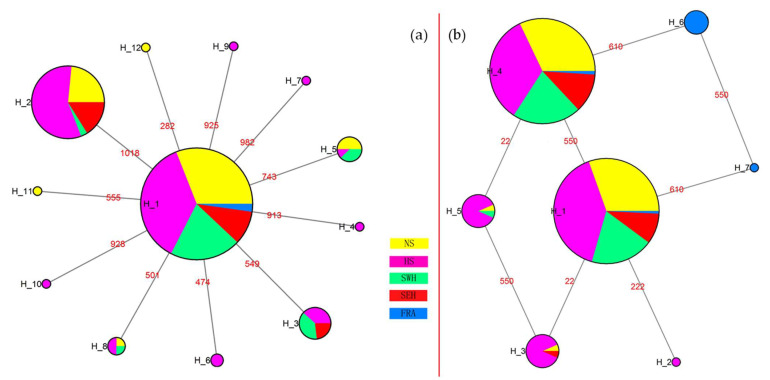
Haplotype network of *R. maidis* in China and France. (**a**): The combined COI and COII genes, H_1 to H_12 represent HRm1 to HRm12; (**b**): the EF-1α gene, H_1 to H_7 represent HRm1 to HRm7. The area of the circles represents the number of individuals sharing the haplotypes. The red numbers indicate the nucleotide positions of polymorphic sites between two haplotypes. NS, north spring maize region in China; HS, Huanghuaihai summer maize region in China; SWH, southwest hilly maize region in China; SEH, southeast hilly maize region in China; FRA, maize region in France.

**Figure 8 insects-14-00057-f008:**
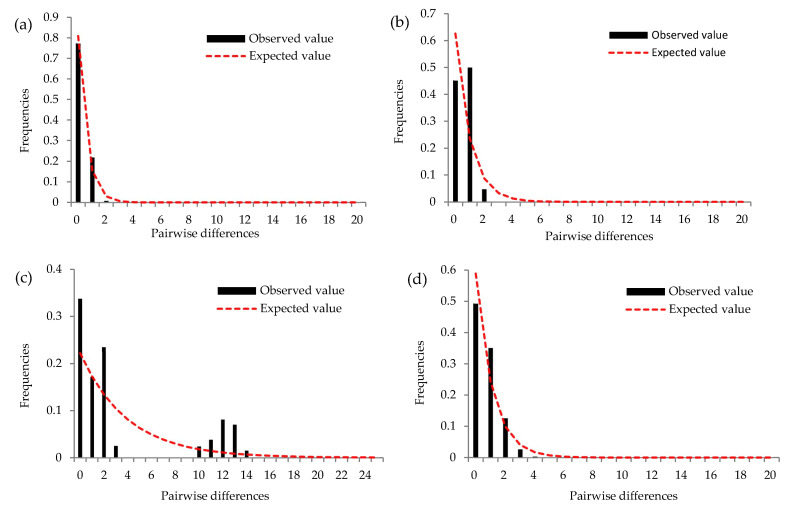
Mismatch distributions of pairwise nucleotide differences between individuals of *R. maidis* and *R. padi*. (**a**) Combined COI-COII gene of *R. maidis*; (**b**) EF-1α gene of *R. maidis*; (**c**) combined COI-COII gene of *R. padi*; (**d**) EF-1α gene of *R. padi*. The numbers of pairwise differences are on the x-axis and their frequencies are on the y-axis.

**Table 1 insects-14-00057-t001:** Analysis of haplotypes and Tajima’s D neutrality test of 32 *R. padi* and 38 *R. maidis* populations.

Aphids	Genes	L (nt)	N	Ha	Hd	S	Pi	D	Fs	Nm
** *R. padi* **	COI/COII	995	535	29	0.663	32	3.52 × 10^−3^	−0.71850 (*p* > 0.10)	−5.305	0.15
	EF-1α	672	535	32	0.507	22	1.04 × 10^−3^	−1.94511 (* *p* < 0.05)	−43.006	1.86
** *R. maidis* **	COI/COII	1065	808	12	0.227	11	0.22 × 10^−3^	−1.75225 (* *p* < 0.05)	−14.460	5.31
	EF-1α	668	808	7	0.548	4	0.89 × 10^−3^	0.12743 (*p* > 0.10)	−1.417	2.89

L, sequence length; N, sequence numbers; Ha, number of haplotypes; Hd, haplotype diversity; S, number of polymorphic sites; Pi, nucleotide diversity; D, Tajima’s D test; Fs, Fu’s Fs test; Nm, overall gene flow estimates; * means there exists a significant difference.

## Data Availability

The data supporting this study’s findings are available from the corresponding author.

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
