# Peer review of "Analysis of the Genetic Diversity of Two Rhopalosiphum Species from China and Europe Based on Nuclear and Mitochondrial Genes"

_insects, 2023, doi:10.3390/insects14010057_

Round 1

Reviewer 1 Report

The authors conducted this study to reveal the influence of geographic distribution on genetic diversity spatial patterns of two species of Rhopalosiphum aphids in corn fields in China and Europe. To describe genetic diversity, they used a nuclear marker and two mitochondrial ones. The results show a relatively low diversity, and a notable differentiation between the samples from Europe and China for R. padi. For R. maidis there was low genetic diversity, and a structure depending on the geographic distribution of the samples was less evident. The migration and constant dispersal of aphids may be the reason for the low structure, except for what was observed in R. padi between regions of China and Europe.

The work is interesting and carefully carried out. It is valuable for the effort to use nuclear and mitochondrial markers with a similarly level of conservation, highly useful to distinguish species.

I have only a couple of general suggestions

(1) Authors could include as much information as possible in the legends of the figures and in the headings of the tables that go in the body of the text, to avoid that the reader must go from one page to another, to fully understand the results. So, repeating the meaning of the legends may be necessary for the benefit of the reader.

(2) The authors apparently collected in regions with different climates, which in turn determined the classification they made of the corn regions (summer, spring,). Is it possible that they are different varieties of maize? Particularly between Europe and China, so that the geographical distribution can include an important effect of the corn variety on which they feed on, in addition to climatic factors. Since Ehrlich and Raven's hypothesis that the formation of races by plants in phytophagous insects is a trigger for speciation is maintained (Janz N. 2011. Annu. Rev. Ecol. Evol. Syst 42: 71-89), perhaps it is worth emphasizing whether this possibility may be involved in the remarkable differentiation that was observed for R. padi, and that the authors even suggest it as a possible subspecies.

In the other hand, I think the simple summary needs minor modifications to make it simpler for a less specialized audience.

Finally, I am attaching a file with suggestions for changes and regarding of the consistency of the use of literals for the identification of parameters, and please be sure of all supplementary tables are available.

Author Response

Dear Dr. Brian T. Forschler,

Thank you very much for the comments sent by Email on our manuscript (ID: insects-2062384). We greatly appreciate three reviewers’ constructive comments and advise that have greatly strengthened our manuscript. Furthermore, the language in this manuscript has been improved by an editing service. We have also carefully made revisions based on the comments by the reviewers. Here is our point-by-point reply to the comments of reviewer 1.

Should you have any questions, suggestions, or need any further information, please let us know.

Please note the lines numbers and figures numbers in our reply are from the revised version. We hope this would be easy for reviewers and editors to follow where the changes were made throughout the manuscript.

Thank you and best regards!

Jianqing Guo

Responses to reviewer 1:

Reviewing:

1) Authors could include as much information as possible in the legends of the figures and in the headings of the tables that go in the body of the text, to avoid that the reader must go from one page to another, to fully understand the results. So, repeating the meaning of the legends may be necessary for the benefit of the reader.

Response: We agree with Reviewer’s comment. The missing information is added in figures 1, 2, 7.

2) The authors apparently collected in regions with different climates, which in turn determined the classification they made of the corn regions (summer, spring,). Is it possible that they are different varieties of maize? Particularly between Europe and China, so that the geographical distribution can include an important effect of the corn variety on which they feed on, in addition to climatic factors. Since Ehrlich and Raven's hypothesis that the formation of races by plants in phytophagous insects is a trigger for speciation is maintained (Janz N. 2011. Rev. Ecol. Evol. Syst 42: 71-89), perhaps it is worth emphasizing whether this possibility may be involved in the remarkable differentiation that was observed for R. padi, and that the authors even suggest it as a possible subspecies.

Response: Thanks for Reviewer’s comment. There may be different varieties of maize in different regions, it’s a pity we did not note the varieties of maize. Thus, we added the description (as follows:) in line 499-501.

“Furthermore, different maize varieties between China and Europe may also contribute to the genetic differentiation of R. padi since the races of host plants act as a trigger for speciation of phytophagous insects [69]”

3) In the other hand, I think the simple summary needs minor modifications to make it simpler for a less specialized audience.

Response: Thanks for Reviewer’s comment. We modified the simple summary combined with the attached file of Reviewer as follows (line 18-30):

Rhopalosiphum padi and Rhopalosiphum maidis are two common phloem-sucking pests on maize which can even transmit viruses (i.e., maize dwarf mosaic virus, MDMV) leading to severe yield losses of maize. Population genetic studies provide information about how much ecological and genetic divergence has been occurred among many near and distant populations. We investigated the genetic diversity of both Rhopalosiphum aphids collected from maize in China and Europe, and we found that different populations of R. maidis showed low genetic variation indicating a high level of gene flow of both nuclear and mitochondrial genes in this aphid. However, mitochondrial gene of R. padi exhibited obvious genetic differentiation between Chinese samples and European samples. In conclusion, the domestic populations of both R. padi and R. maidis showed low genetic diversity and the long distance between China and Europe may interrupt the gene exchange of aphid.”

4) Finally, I am attaching a file with suggestions for changes and regarding of the consistency of the use of literals for the identification of parameters, and please be sure of all supplementary tables are available.

Response: Thanks for Reviewer’s comment. We revised the manuscript based on all the comments in the attached file, and the responses are as follows.

Comments in attached file:

1) Is there a missing name?

Response: Thanks for Reviewer’s kind comment. The word “and” should between the fifth and the sixth authors, and we corrected (line 5).

2) Comments for simple summary

Response: Thanks for Reviewer’s kind comment. We revised the simple summary as follows (line 18-30):

Rhopalosiphum padi and Rhopalosiphum maidis are two common phloem-sucking pests on maize which can even transmit viruses (i.e., maize dwarf mosaic virus, MDMV) leading to severe yield losses of maize. Population genetic studies provide information about how much ecological and genetic divergence has been occurred among many near and distant populations. We investigated the genetic diversity of both Rhopalosiphum aphids collected from maize in China and Europe, and we found that different populations of R. maidis showed low genetic variation indicating a high level of gene flow of both nuclear and mitochondrial genes in this aphid. However, mitochondrial gene of R. padi exhibited obvious genetic differentiation between Chinese samples and European samples. In conclusion, the domestic populations of both R. padi and R. maidis showed low genetic diversity and the long distance between China and Europe may interrupt the gene exchange of aphid.”

3) Comments for Abstract

Response: Thanks for Reviewer’s comment. We revised the Abstract (combined with other reviewers’ comments) as follows (line 31-49):

“Population genetic studies can reveal clues about the evolution of adaptive strategies of aphid species in agroecosystems and demonstrate the influence of environmental factors on the genetic diversity and gene flow among aphid populations. To investigate the genetic diversity of two Rhopalosiphum aphid species from different geographical regions, 32 populations (n=535) of the bird cherry-oat aphid (Rhopalosiphum padi Linnaeus) and 38 populations (n=808) of the corn leaf aphid (Rhopalosiphum maidis Fitch) from China and Europe were analyzed using one nuclear (elongation factor-1 alpha) and two mitochondrial (cytochrome oxidase I and II) genes. Based on the COI-COII sequencing, two obvious clades between Chinese and European populations and low level of gene flow (Nm=0.15) were detected in R. padi while no geographical-associated genetic variation was found for EF-1α in this species. All genes in R. maidis had low genetic variation, indicating a high level of gene flow (Nm=5.31 of COI-COII and Nm=2.89 of EF-1α). Based on the mitochondrial result of R. padi, we concluded that the long distance between China and Europe may be interrupting the gene flow. The discordant results of nuclear gene analyses in R. padi may be due to the slower evolution of nuclear genes compared to mitochondrial genes. The gene exchange may occur gradually with the potential of continuous migration of the aphid. This study facilitates the design of control strategies for these pests.”

4) I could not have acces to supplementary TS1-TS7, may be was my problem. Please check the information is availably

Response: Thanks for Reviewer’s comment. We made a mistake for uploading the supplementary materials. Tables S1–S7 are in a Word document and I will upload it this time during the revision.

5) I Suggest this "unity" within text, in some part the authors use "nt"

Response: Thanks for Reviewer’s comment. We unified the unit with “nt” throughout the text.

6) There is some confusion between use variant o regions

Response: Thanks for Reviewer’s comment. We deleted “variant” during the language editing (line 268).

7) I suggest using three decimal places.

Response: We agree with Reviewer’s comment and corrected the data (line 285, 287, 297-299, 307-308, 313-314, 333-334).

8) The authors use a slightly different notation in the text, I suggest they harmonize it. (Table 2)

Response: Thanks for Reviewer’s comment. We harmonized the format throughout the text as well as the supplementary materials.

9) It is recommended to use only three decimal places. (Table 2)

Response: We agree with Reviewer’s comment and corrected the data. The original Tables 2 and 3 have been moved to supplementary materials as Tables S8 and S9 which was recommended by another Reviewer.

10) I suggest that the haplotype clustering and network be presented within a same figure, for each molecular marker and by species. It is confusing that there is no sequential numbering during the call to see figures of results, also what I mentioned before happens that moving to see results distracts the reading.

Response: Thanks for Reviewer’s comment. Based on all Reviewers’ comments, we changed the original figures 3 and 4 to figure 4a and 4b, the original figures 5 and 6 to figure 5a and 5b, the original figures 7 and 8 to figure 6a and 6b, and the original figures 9 and 10 to figure 7a and 7b. We hope that it will be easier for reading this time.

11) 400-403 This statement needs a reference.

Response: Thanks for Reviewer’s comment. We added the following reference (line 522): 

“Gupta, B; Kaur, J. Computational analysis of conserved coil functional residues in the mitochondrial genomic sequences of dermatophytes[J]. Bioinformation 2016, 12, 197-201.”

12) How does the absence of sexual reproduction affect to the levels of diversity and structure of Rhopalosiphum spp? (line 403-404)

Response: Thanks for Reviewer’s comment. Literatures (below) reported that asexual populations showed less allelic polymorphism compared with sexual populations, which may contribute to the low genetic diversity of R. maidis in our study and we added the discussion as follows (line 511-514):

“Obligate parthenogenesis of R. maidis reduces the chance of gene recombination and the parthenogenetic populations show less allelic polymorphism compared with sexual reproduction [38], which may contribute to the low genetic diversity.”

Literatures:

“Delmotte, F.; Leterme, N.; Gauthier, J.P.; Rispe, C.; Simon, J.C. Genetic architecture of sexual and asexual populations of the aphid Rhopalosiphum padi based on allozyme and microsatellite markers. Mol. Ecol. 2002, 11, 711-723.”

“Kanbe, T.; Akimoto, S.I. Allelic and genotypic diversity in long-term asexual populations of the pea aphid, Acyrthosiphon pisum in comparison with sexual populations[J]. 2009, 18, 801-816.”

13) America

Response: Thanks very much, corrected (line 578).

14) The authors apparently collected in regions with different climates, which in turn determined the classification they made of the corn crop (summer, spring,..). Is it possible that they are different varieties of corn? in particular between Europe and China, so that the geographical can include an important selective effect of the variety on which they feed, in addition to climatic factors. Since Herlich and Raven's hypothesis (Janz N. 2011. Annu. Rev. Ecol. Evol. Syst 42: 71-89) that the formation of races by plants in phytophagous insects is a trigger for speciation is maintained, perhaps it is worth emphasizing whether this possibility may be involved in the remarkable differentiation that was observed for padi, and that the authors even suggest it as a possible subspecies.

Response: Thanks for Reviewer’s comment. There may be different varieties of maize in different regions, it’s a pity we did not note the varieties of maize. Thus, we added the description (as follows:) in line 499-501.

“Furthermore, different maize varieties between China and Europe may also contribute to the genetic differentiation of R. padi since the races of host plants act as a trigger for speciation of phytophagous insects [69]”

15) Even the low diversity can be due to the type of markers, both are preferentially chosen to separate species, so they are markers with low evolutionary rates, so it is expected that within each species they will be little or no variable. However, the authors found at least in padi an interesting differentiation between European and Chinese regions.

Response: We agree with Reviewer’s comment and deleted the sentence.

Reviewer 2 Report

Rhopalosiphum padi and R. maidis are damaging pests with a worldwide distribution. Studies of population genetics can help improve our understanding of the eco-evolutionary dynamics of these aphids, with implications for pest management. In their manuscript, the authors offer an excellent study of the population genetics of Chinese and European Rhopalosiphum aphids (the first to focus on maize as the host plant) and with a high level of resolution for populations within the four major maize-growing regions in China. For my review, I will limit my comments to the population ecology and dynamics of aphids. Regarding the analysis of population genetics, while the Methods/Results sections are clearly written and appear scientifically sound, I am less familiar with this type of research and will defer to the expertise of other reviewer(s). Overall, I only have a few minor comments. I did not comment on grammar/syntax in my review, but the manuscript could be further improved with edits for English.

DETAILED COMMENTS:

L. 19: Spell out acronyms the first time they are used (maize dwarf mosaic virus, MDMV).

L. 28: This is a good point; I would like you to expand the discussion to include a paragraph discussing how the regional environment may affect the population genetics/dynamics of the aphids (the four growing regions in China and the region in Europe).

L. 37–38: I don’t understand this—the present study did not monitor population densities over time.

L. 42–43: You should delete this or expand on this idea in the discussion. Landscape- or geographic-level control strategies may be difficult to implement but are certainly an interesting possibility.

L. 82: “allozyme” is not capitalized

Tables S1–S7 are missing from my copy of the supplemental materials. I only have Tables S8–S17. I would need to see these other tables before making a recommendation for publication.

L. 135: “R. maidis

Table 1: What do values in bold indicate?

L. 235–237: How many significant figures do you have here?

Fig. 4: What does the orange color for HRm7 represent?

Figs. 5 and 6: Aphis glycines is listed as the outgroup insect, but the label for the outgroup is Schizaphis graminum. Also, why are you using a different outgroup species here?

L. 352: “Tajima’s D”

Fig. 11c: What is this second peak around 12 pairwise differences? How do you interpret that?

L. 403: Except that, as you state in L. 76, “R. maidis generally lack sexual reproduction,” and very little (if any) genetic recombination would be expected through parthenogenesis. Maybe you can also compare/contrast your observed genetic differences based on differences in reproduction/cyclic parthenogenesis in the two species.

Author Response

Dear Dr. Brian T. Forschler,

Thank you very much for the comments sent by Email on our manuscript (ID: insects-2062384). We greatly appreciate three reviewers’ constructive comments and advise that have greatly strengthened our manuscript. Furthermore, the language in this manuscript has been improved by an editing service. We have also carefully made revisions based on the comments by the reviewers. Here is our point-by-point reply to the comments of reviewer 2.

Should you have any questions, suggestions, or need any further information, please let us know.

Please note the lines numbers and figures numbers in our reply are from the revised version. We hope this would be easy for reviewers and editors to follow where the changes were made throughout the manuscript.

Thank you and best regards!

Jianqing Guo

Responses to reviewer 2:

Reviewing:

1) Comments and Suggestions for Authors

Rhopalosiphum padi and R. maidis are damaging pests with a worldwide distribution. Studies of population genetics can help improve our understanding of the eco-evolutionary dynamics of these aphids, with implications for pest management. In their manuscript, the authors offer an excellent study of the population genetics of Chinese and European Rhopalosiphum aphids (the first to focus on maize as the host plant) and with a high level of resolution for populations within the four major maize-growing regions in China. For my review, I will limit my comments to the population ecology and dynamics of aphids. Regarding the analysis of population genetics, while the Methods/Results sections are clearly written and appear scientifically sound, I am less familiar with this type of research and will defer to the expertise of other reviewer(s). Overall, I only have a few minor comments. I did not comment on grammar/syntax in my review, but the manuscript could be further improved with edits for English.

Response: Thanks very much for Reviewer’s comment. The language in this manuscript has been improved by one of the editing services.

DETAILED COMMENTS:

2) 19: Spell out acronyms the first time they are used (maize dwarf mosaic virus, MDMV).

Response: Thanks for Reviewer’s kind comment, we added (line 19).

3) 28: This is a good point; I would like you to expand the discussion to include a paragraph discussing how the regional environment may affect the population genetics/dynamics of the aphids (the four growing regions in China and the region in Europe).

Response: Thanks for Reviewer’s comment, we modified this part in the discussion as follow (line 531-556):

“The two distinct clades of the combined COI-COII (Figure 6a) in R. padi revealed that the substantial geographical distance between China and European countries may result in distinct aphid lineages (i.e., subspecies) based on geographical differentiation. By contrast, low genetic diversity has often been detected in insects collected from a small geographical range (e.g., within a country) [50, 76–78]. However, most specimens from the HS region were distributed in HRp1, and more than half of the specimens from the SWH region and half from the NWI region were distributed in HRp2 and HRp3, respectively, of the mtDNA in R. padi. This result indicates that there exists some genetic differentiation within China despite the frequent gene exchange. Besides, the sampling sites in Europe (temperate maritime climate) have mild winters whereas NWI region (temperate continental area with arid climate) and HS region (temperate monsoon climate with high temperature and rainy summer) have cold winters, which may affect the population genetic diversity of R. padi [41]. Regarding to the European clade, we detected eight haplotypes not present in Chinese populations. Nevertheless, no obvious genetic diversity was found among the eight haplotypes, which may be the result of the few collection sites (four sites) from European region because the longitudinal cline related distribution of R. padi in France was described in a previous study [43]. In addition, three out of 29 R. padi from the HRp6 (in the European clade) of mtDNA were collected from the NWI region of China. We speculate that three possibilities may explain this phenomenon: (1) European R. padi might have been passively carried to the NWI region through agricultural products and then colonized the region; (2) A spontaneous mutation occurred in Chinese R. padi which resulted in HRp6; (3) Gene exchange might have occurred indirectly through continuous migration across countries between China and Europe, which needs further research. By contrast, the genetic diversity of EF-1α in R. padi was low (Figure 6b).”

4) 37–38: I don’t understand this—the present study did not monitor population densities over time.

Response: Thanks for Reviewer’s comment, we deleted the sentence.

5) 42–43: You should delete this or expand on this idea in the discussion. Landscape- or geographic-level control strategies may be difficult to implement but are certainly an interesting possibility.

Response: Thanks for Reviewer’s comment, we deleted the part of “such as controlling the migration of these aphids” (line 42).

6) 82: “allozyme” is not capitalized

Response: Thanks, we corrected (line 104).

7) Tables S1–S7 are missing from my copy of the supplemental materials. I only have Tables S8–S17. I would need to see these other tables before making a recommendation for publication.

Response: Thanks very much for Reviewer’s comment, we made a mistake for uploading the supplementary materials. Tables S1–S7 are in a Word document and I will upload it this time during the revision.

8) 135: “R. maidis

Response: Thanks very much for Reviewer’s comment, we corrected (line 165).

9) Table 1: What do values in bold indicate?

Response: Thanks for Reviewer’s kind comment. We removed the bold since there is no special value.

10) 235–237: How many significant figures do you have here?

Response: Thanks for Reviewer’s comment. We rewrote the sentence (as follow) since there is no significant difference (line 284-288).

“Moreover, the genetic distance (FST) of R. padi (0.167 for EF-1α and 0.381 for the combined COI-COII) has no significant difference with R. maidis (0.046 for EF-1α and 0.065 for the combined COI-COII).”

11) 4: What does the orange color for HRm7 represent?

Response: Thanks for Reviewer’s kind comment. Maybe the color of HRm7 was neglected during the layout of the manuscript. There are two colors in Fig. 5a (revised figure number) and we corrected the color.

12) 5 and 6: Aphis glycines is listed as the outgroup insect, but the label for the outgroup is Schizaphis graminum. Also, why are you using a different outgroup species here?

Response: Thanks for Reviewer’s kind comment. We should keep consistent for the outgroup and we modified the figures but forgot to revise the captions. We corrected the captions of Fig. 5 (original fig 5 and 6) this time.

13) 352: “Tajima’s D”

Response: Thanks for Reviewer’s kind comment. We corrected (line 449).

14) 11c: What is this second peak around 12 pairwise differences? How do you interpret that?

Response: Thanks for Reviewer’s comment. The value of Tajima’s D was not significant (Table 1) for combined COI-COII gene of R. padi (this figure), combined with the third Reviewer’s comment, we rewrote this part (line 449-455).

15) 403: Except that, as you state in L. 76, “R. maidis generally lack sexual reproduction,” and very little (if any) genetic recombination would be expected through parthenogenesis. Maybe you can also compare/contrast your observed genetic differences based on differences in reproduction/cyclic parthenogenesis in the two species.

Response: We accept Reviewer’s comment. The discussion about the reproduction/cyclic parthenogenesis of R. maidis was added in line 511-514 and the discussion for R. padi is in line 562-569.

line 511-514:

“Obligate parthenogenesis of R. maidis reduces the chance of gene recombination and the parthenogenetic populations show less allelic polymorphism compared with sexual reproduction [38], which may contribute to the low genetic diversity.”

line 562-569:

“The life cycle can also affect the gene mutations in aphids. A study using mtDNA as a marker demonstrated that R. padi has an incomplete life cycle (i.e., exists as an obligate parthenogenetic population) and shows the haplotype I, whereas cyclical parthenogenesis populations exhibit distinct haplotypes [16]. Likewise, a microsatellite marker revealed substantial genetic differences between obligate parthenogenetic and cyclically parthenogenetic populations of R. padi collected from wheat [26].”

Reviewer 3 Report

This manuscript presents DNA sequence data for two aphid species with good sampling across China and a few European samples. Variation at two loci was analysed using suitable tools. The opportunity to contrast two species is valuable. The failure to detect Chinese mtDNA haplotypes in European samples of R. padi does suggest a partial barrier to gene flow. Unfortunately, the context to the data is poorly explained, important details are missing and the assumptions made when estimating gene flow are not stated. Additional analyses and improvements to the writing are needed.    

Aphids that feed on maize are very likely to have become more abundant and widespread over the last 9000 years since their host plant was domesticated and introduced from Mexico to the world. Population expansion will leave a genetic signature if it occurred relatively recently. But previous studies of these aphids suggest their reproductive mode is linked to the level of genetic structure in R. padi – with more geographic differentiation detected in the obligate parthenogens. As R. maidis is mostly parthenogenetic it too might show genetic structure.  Introduction of the manuscript should explain which reproductive mode occurs in these two aphid species in China – if not known then perhaps explain why this is not known.

Do these two aphid species feed on other plants in China?  If estimating gene flow it will help to know whether other populations exist within the region. 

Introduction/aims should set out what the authors expected to find given the history of the host plants and the reproductive mode of the aphids in China.

1.     If the aphids have expanded rapidly (and can disperse easily (lines 403-406)) one would not expect to see geographic structure nor isolation by distance. 

2.     However, multiple introductions, selection or host plant differentiation might result in genetic structure. Only if one assumes an equilibrium between migration and drift, can gene flow be estimated. 

Analyses should include

1.     Table of pairwise distances for each species could be in the main text (could present mtDNA (phi)ST and FST estimated from EF1-a in same table) – evidence of a departure from homogenous populations has already been tested using significant deviations from zero – however it is appropriate to use a Bonferroni correction for multiple tests. Low (Phi)ST and Fst values as seen in the current supplementary data suggest little population structure in the two aphid species.

2.     Isolation by distance test (pairwise PhiST and FST against geographic distance) would help establish whether structure results from stable population with gene flow or just recent range expansion.

Minor:

1.     How many copies of EF-1α in aphids? Beetles, bees and flies have two, moths have one. Aphids? 

2      I assume the authors estimated (phi)ST from mtDNA but did they provide reference of this measure (which differs from FST)?

3.     Estimating gene flow would require assumptions about population size and structure -island model, equilibrium between migration and drift, no selection, stable population size, all sub-populations are the same size, (see Whitlock & McCauley 1999 Indirect measures of gene flow and migration: FST≠1/(4Nm+1)). These assumption need to be stated in the text. 

4.     The authors should look for a signature of isolation by distance. Gene flow is likely to be limited by distance (assuming the loss of alleles by drift and their replacement by migration are in equilibrium, and that dispersal rates are constant across the geographical range).  

5.     Given that COI and COII are part of the same locus I think that only the combined data COI + COII should be used for the tests shown in table 1. 

6.     Stable population or expanding? – Given the historical expansion of host plant one would expect these two aphids would also have expanded. Tajima’s D is significant for EF1-a from R. padi, and mtDNA for R. maidis (*P < 0.05). Thus two of the four conservative tests seem to provide evidence of population growth. Most are negative and might lack power.  The shape of the network mtDNA R. maidis (figure 9) suggests population of this species are expanding.  If aphid populations have (or are) expanding then gene flow estimates will not be reliable. 

7.     The term “dominant” would be better replaced with ‘common’ or ‘abundant’ (line 18; 65)

8.     Add COI sequence data from previous studies of these aphid species into figures 3 & 5 (e.g. references 18 20? 38?) or explain why you cannot.

9.     I don’t think the AMOVA results add anything to the paper – consider moving these to supplementary. I agree with the conclusions expressed in lines 392-397.

10.  Figures: can fig 3 and 4 be shown side by side (a & b)?  Can fig 5 and 6 be shown side by side?

11.  English can be understood but has many grammatical errors and occasionally incorrect use of jargon. Here is an example (changes in bold): 

Intraspecific genetic diversity provides the basis for investigation of the evolutionary changes history of species as well as offersing the opportunity to document the basic level of biodiversity within and among populations [1]. Hence, population genetics has highlighted the importance has been highlighted to of studing the molecular variability within species. In addition to different life cycles of aphids [2] and specialisation to host plant species [3-4], geographical isolation of aphid populations may as well generate genetic structure allopatric speciation resulting from both drift and selection in different environmental conditions [5].
